# Accuracy of Fetal Biacromial Diameter and Derived Ultrasonographic Parameters to Predict Shoulder Dystocia: A Prospective Observational Study

**DOI:** 10.3390/ijerph19095747

**Published:** 2022-05-09

**Authors:** Marco La Verde, Pasquale De Franciscis, Clelia Torre, Angela Celardo, Giulia Grassini, Rossella Papa, Stefano Cianci, Carlo Capristo, Maddalena Morlando, Gaetano Riemma

**Affiliations:** 1Obstetrics and Gynecology Unit, Department of Woman, Child and General and Specialized Surgery, University of Campania “Luigi Vanvitelli”, 80128 Naples, Italy; marco.laverde88@gmail.com (M.L.V.); pasquale.defranciscis@unicampania.it (P.D.F.); clelia.t@hotmail.it (C.T.); angelacelardo@gmail.com (A.C.); giulia.grassini9@gmail.com (G.G.); rossellapapa@gmail.com (R.P.); madmorlando@gmail.com (M.M.); gaetano.riemma7@gmail.com (G.R.); 2Unit of Gynecology and Obstetrics, Department of Human Pathology of Adult and Childhood “G. Barresi”, University of Messina, 98122 Messina, Italy; 3Pediatrics Unit, Department of Woman, Child and General and Specialized Surgery, University of Campania “Luigi Vanvitelli”, 80128 Naples, Italy; carlo.capristo@unicampania.it

**Keywords:** shoulder dystocia, fetal biacromial diameter, ultrasound fetal biacromial diameter, fetal ultrasound biometry, delivery, fetal macrosomia

## Abstract

Background and Objectives: Shoulder dystocia (ShD) is one of most dangerous obstetric complication. The objective of this study was to determine if the ultrasonographic fetal biacromial diameter (BA) and derived parameters could predict ShD in uncomplicated term pregnancies. Materials and Methods: We conducted a prospective observational study in a tertiary care university hospital from March 2021 to February 2022. We included all full-term pregnancies accepted for delivery that received an accurate ultrasonography (USG) scan before delivery. USG biometry and estimated fetal weight (EFW) were collected. Therefore, we evaluated the diameter of the mid-arm, the transverse thoracic diameter (TTD) and the biacromial diameter (BA). BA was estimated using Youssef’s formula: TTD + 2 mid-arm diameters. The primary outcome was the evaluation of BA and its related parameters (BA/biparietal diameter (BPD), BA/head circumference (HC) and BA–BPD in fetuses with ShD versus fetuses without ShD. Diagnostic accuracy for ShD of BA, BA/BPD, BA/HC and BA–BPD was evaluated using receiver operator curve (ROC) analysis. Results: 90 women were included in the analysis, four of these had ShD and required extra maneuvers after head delivery. BA was increased in fetuses with ShD (150.4 cm; 95% CI 133.2 cm to 167.6 cm) compared to no-ShD (133.5 cm; 95% CI 130.1 cm to 137.0 cm; *p* = 0.04). Significant differences were also found between ShD and no-ShD groups for BA/BPD (1.66 (95% CI 1.46 to 1.86) vs. 1.44 (95% CI 1.41 to 1.48); *p* = 0.04), BA/HC (0.45 (95% CI 0.40 to 0.49) vs. 0.39 (95% CI 0.38 to 0.40); *p* = 0.01), BA–BPD (60.0 mm (95% CI 42.4 to 77.6 cm) vs. 41.4 (95% CI 38.2 to 44.6); *p* = 0.03), respectively. ROC analysis showed an overall good accuracy for ShD, with an AUC of 0.821 (*p* = 0.001) for BA alone and 0.881 (*p* = 0.001), 0.857 (*p* = 0.016) and 0.867 (*p* = 0.013) for BA/BPD, BA–BPD and BA/HC, respectively. Conclusions: BA alone, as well as BA/BPD, BA/HC and BA–BPD might be useful predictors of ShD in uncomplicated term pregnancies. However, such evidence needs extensive confirmation by means of additional studies with large sample sizes, especially in case of pregnancies at high risk for ShD (i.e., gestational diabetes).

## 1. Introduction

Shoulder dystocia (ShD) represents an extremely harmful obstetric complication. It is described as the inability to deliver the fetus’ shoulders after head delivery. Even if all recommended measures are adopted, there is a high risk of catastrophic neonatal and maternal morbidity and mortality [1,2,3]. Multiple risk factors for shoulder dystocia were investigated. The link between fetal macrosomia and ShD was established [4], and the American College of Obstetricians and Gynaecologists recommends a cesarean delivery (CD) to prevent ShD in diabetic women with an estimated fetal weight (EFW) over the 4500 g and in non-diabetic women who had an EFW of at least 5000 g [5]. Nevertheless, most of ShD cases (50%) happen without an EFW over these cut-offs [6]. Other potential risk factors evaluated were: instrumental birth [7,8], high BMI [9], chronic diabetes [10] and gestational diabetes [11]. Besides these well-established risk factors, other risk variables suggested in the scientific work include non-white race/ethnicity [12], shorter height [13], multiparity [14], history of past shoulder dystocia [15], prolonged labor [16] and labor induction [17] despite the type of induction [18,19,20]. Different types of anesthesia are practiced in obstetricians’ routine [21,22,23], however, some evidence suggests that epidural anesthesia could raise the incidence of shoulder dystocia [24]. Despite all these risk factor evaluations, ShD remains unpredictable [5]. According to a large retrospective cohort study, sonographic fetal anthropometric measurements do not appear a helpful technique for ShD screening, with a low positive predictive value [11]. Other authors discovered that ShD was expected when the fetal shoulder’s mean diameter was prominent [25,26]. Based on these assumptions, Youssef et al. suggested a new ultrasound fetal biometrics measure to predict the fetal biacromial diameter (BA) and reported a correlation between the BA and fetal macrosomia [27]. On these premises, we used Youssef’s formula to calculate the intrauterine fetal biacromial diameter and its potential association with ShD onset.

## 2. Materials and Methods

We performed a single-center prospective observational study at a tertiary-care university hospital from March 2021 to February 2022. Our Institutional Ethical Review Board had authorized the research design (protocol no. 0013626/i date: 3 May 2021), and all participating women submitted written consent after a detailed discussion. We included all full-term pregnancies (37–42 weeks) accepted for the delivery. We excluded pregnancies with antepartum hemorrhage, fetal growth restriction, stillbirths, multiple gestations, breech position, uterine or congenital malformations. In addition, the complete history with age, parity, past miscarriages, body mass index (BMI) and any medical illnesses was recorded for each patient. The fetal diameters were assessed utilizing a Voluson E8 (General Electric Medical Systems, Zipf, Austria) or a Samsung HS70 (Samsung Medison, Seoul, South Korea) with a transabdominal convex probe (3.5–5.5 MHz). To reduce the bias, all ultrasound scans were conducted only by two expert sonographers (M.L.V. and M.M.) (equipped with the Fetal Medicine Foundation certificate of competence in the 11–13-week scan) who performed all the ultrasonographic examinations of enrolled women [28]. We recorded the main fetal ultrasonographic parameters: biparietal diameter (BPD), head circumference (HC), abdominal circumference (AC), femur length (FL) and amniotic fluid index (AFI). Estimated fetal weight (EFW) was calculated using the Hadlock 4 formula [29]. We also registered two additional parameters: the mid-arm diameter measured by placing the caliper from skin to skin on the upper arm near the heart (Figure 1) and the transverse thoracic diameter (TTD) estimated by determining a transverse section of the fetal chest at the level of the heart (four-chamber view) (Figure 2). 

After this, we calculated the fetal BA, applying Youssef’s ultrasonography formula: [TTD + 2 mid-arm diameter] [27]. All pregnant women underwent an ultrasound exam 24–48 h before the delivery. All women had an admission computerized cardiotocography (CTG) and were monitored with CTG during labor according to hospital guidelines [30,31,32,33]. After the birth, the following data were collected: delivery method, neonatal birth weight, Apgar scores at 1 and 5 min, presence of any difficulties during the delivery and occurrence of ShD maneuvers during the vaginal delivery. The primary endpoint was the comparison of ultrasound fetal BA in fetuses with ShD versus the no-ShD group (control group). The ShD was defined as the need for extra obstetric procedures to deliver the baby after the head was delivered and mild traction was unsuccessful [34]. We considered the BA combined with other ultrasonographic parameters in the ShD group as secondary endpoints: BA/BPD; BA/HC, BA–BPD. 

Based on the current literature, an a priori calculation for the size of sample required to detect a minimum number of ShDs was performed: given 80% power and an alpha level of 0.05, and an estimated incidence of 0.3%, considering a 7% loss to follow up, a minimum of 84 women were needed to detect significant differences in the primary outcome of interest.

The data were coded, tabulated and analyzed using GraphPad PRISM 9 (GraphPad, La Jolla, CA, USA) The Student’s t-test was used to assess quantitative data. The Mann–Whitney test was used to examine qualitative data. Receiver operating characteristic (ROC) curves were calculated to determine the predictive value of the BA and surrogate parameters to predict ShD at birth. Therefore, sensitivity, specificity and positive likelihood ratio were obtained for each ultrasonographic parameter. A *p*-value (*p*) lower or equal to 0.05 was considered statistically significant.

## 3. Results

Ninety pregnant women were prospectively enrolled. During labor, four of these patients needed additional maneuvers after head delivery (McRoberts’ and Rubin’s maneuvers were performed). Table 1 described the demographic and baseline characteristics of the included patients. Eleven patients were affected by gestational diabetes (12.2%), one had pregestational diabetes (1.1%) and eight women developed gestational hypertension and preeclampsia (8.8%) (Table 1).

Table 2 displays the ultrasound measures evaluated and EFW estimated before labor. The ShD group had an increased BA with a mean of 150.4 cm (95% CI 133.2 cm to 167.6 cm) versus the 133.5 cm (95% CI 130.1 cm to 137.0 cm) of the control group (*p*-value 0.04). In addition, the BA/BPD, BA/HC ratio and BA–BPD showed statistical significance with a higher mean in the ShD group compared to non-ShD group (Table 2). On the other hand, the BA, maternal height ratio and the EFW did not show a statistical difference among the two groups (Table 2).

The ROC curve analysis showed that BA has a good diagnostic accuracy for ShD, with an AUC of 0.821 (0.735 to 0.906; *p* = 0.001), with a threshold > 138.3 mm, showing a sensitivity of 100.0 (95% CI 75.75% to 100.0%), a specificity of 68.60% (95% CI 58.18% to 77.44%) and a positive likelihood ratio of 3.185 (Figure 3).

The BA/BPD ratio also showed a good diagnostic accuracy for predicting ShD, with an AUC of 0.881 (0.784 to 0.977; *p* = 0.001), showing the highest positive likelihood ratio of 5.250 when the threshold was set >1.625, with a sensitivity of 75.00% (95% CI 30.06% to 98.72%) and a specificity of 85.71% (95% CI 76.67% to 91.63%) (Figure 4).

The difference between BA and BPD had an AUC of 0.857 (95%, CI 0.750 to 0.963; *p* = 0.016), with a positive likelihood ratio of 4.446 at a threshold of at least 54.8 mm, with a 75.00% sensitivity (9%5 CI 30.06% to 98.72%) and 83.13% specificity (95% CI 73.66% to 89.68%) (Figure 5).

Similarly, BA/HC ratio showed an AUC of 0.867 (95% CI 0.773 to 0.962; *p* = 0.013), showing that a threshold of at least 0.435 was the best cut-off, with a 4.500 positive likelihood ratio, a sensitivity of 75.00% (95%, CI 30.06% to 98.72%) and a specificity of 83.33% (95 CI 73.95% to 89.80%) (Figure 6).

Seventy-eight women (86.67%) had a vaginal delivery with an 11.11% rate of need for operative delivery. Twelve pregnant women (13.33%) delivered by cesarean section. Other delivery data, including Apgar scores, mean birth weight, birth length and mean cranial circumference are shown in Table 3.

## 4. Discussion

Shoulder dystocia is an uncommon event yet has the potential to cause severe neonatal injury. Prediction is challenged by uncertain factors such as maternal pelvic size or fetal body [8,35]. Numerous risk factors influence the development of ShD and different researchers have employed various risk factor combinations to find a predictive model for ShD [11,16,34,36,37,38]. In modern obstetrics, managing ShD is difficult. The retrospective methodology of available studies, as well as the nonuniform criteria of both macrosomia and ShD and the lack of randomization, limit their findings. The function of ultrasonographic evaluation in the definition, diagnosis and management of ShD has been a point of contention so far [39]. Dyachenko et al. analyzed a statistical model for detecting ShD with brachial plexus injury and discovered a model based on maternal height and weight, gestational age and parity, which identified 50.7% of the total cases [40]. Belfort et al. performed a multivariate statistical analysis on patients with ShD and evidenced the association between the birthweight, operative vaginal delivery and ShD [36]. Ultrasound imaging has been ineffective in identifying ShD cases to prophylactically plan for cesarean delivery [41]. Miller et al. evaluated a targeted ultrasound strategy based on abdominal diameter minus biparietal diameter to predict the ShD, although the predictive values and false-positive rates are not favorable [42]. Gerber et al. discovered that an abdominal/head circumference ratio of more than 1.05 had a sensitivity and specificity of 46 and 75%, respectively, with a positive predictive value of only 5.7% [43]. Other authors reported comparable results [11]. Youssef et al. evidenced the association between a new ultrasound formula (to measure the fetal biacromial diameter), fetal macrosomia and ShD, which was the starting point of our trial [27]. Similarly, Terzi et al. evaluated the role of fetal clavicle length to predict ShD [44]. Their study showed a comparable diagnostic accuracy relative to BA. However, their study was also limited by a reduced sample size [44]. In our study, the four women who experienced ShD were heterogenous, with two of them with a diagnosis of gestational diabetes mellitus. Two of them were nulliparous while the other two had delivered twice before. Height ranged from 160 to 171 cm while weight ranged from 66 to 81 kg, with an estimated BMI that ranged from 27.1 to 31.0 kg/m^2^. Regarding the differences between diabetic and non-diabetic women, there were no significant changes relative to ultrasonographic fetal biometric parameters among the cases. However, due to the extreme rarity of the ShD, which led only to the four included cases, it was not plausible to perform a regression analysis for the abovementioned covariates. Previous key research attempting to develop a model for predicting ShD based on different risk-factor combinations, with and without sonographic EFW, failed to match the objective. A machine learning model, on the other hand, could add increased precision to previously observed algorithms by combining multiple characteristics of both the woman and the fetus, as described by Tsur et al. [38]. However, to date, no one method is proper and validated for preventing ShD [45]. Our study evidenced a statistically significant difference in ultrasonography BA, BA/BPD, BA/HC and BA–BPD in the group of pregnant women who experienced shoulder delivery difficulty at birth. According to the ROC curve analysis, BA and BA/BPD ratio had high diagnostic accuracy for ShD. Similarly, BA and BPD difference and BA/HC ratio had a significant AUC. Actually, ShD occurs as an unpredictable obstetric emergency and the obstetric staff follows multiple maneuvers to resolve the dystocia [46,47,48]. Due to the general low sensitivity and specificity, no previous ultrasonographic formula was employed in a clinical protocol. The formula presented here could help the early detection of ShD cases and support clinical decision-making. However, due to the rare incidence of ShD, more data are needed before its implementation in clinical practice. Therefore, our results suggest that a large multicentric prospective study based on our findings would not be unrealistic, especially for the ShD high-risk population of diabetic pregnancies [49,50,51]. Nonetheless, other authors have tried to improve the prediction of ShD by analyzing other ultrasonographic parameters, including fetal abdominal and thigh soft tissue thickness [39,52]. BA and its surrogate parameters seem to have increased diagnostic accuracy; however, comparative trials are needed to evaluate differences regarding the various ultrasonographic measures available. The strengths of our study include its prospective cohort design and the adoption of the abovementioned and previously validated formula (Youssef’s formula). In addition, we tested new ultrasonographic parameters and assessed their diagnostic accuracy for the first time (BA/BPD, BA/HC and BA–BPD). The study’s main limitation was the rare ShD incidence that needs a broader sample size to further validate the actual findings.

## 5. Conclusions

Ultrasonographic evaluation of the fetal biacromial diameter appears to be predictive for shoulder dystocia in an unselected cohort of pregnant women at term. Therefore, detailed comprehension of pertinent pelvic and fetal anatomy and the mechanisms behind dystocia onset should be further clarified. For this reason, larger-scale prospective studies are required to assess the shoulder dystocia predictive capability of the ultrasound biacromial diameter and its surrogate parameters.

## Figures and Tables

**Figure 1 ijerph-19-05747-f001:**
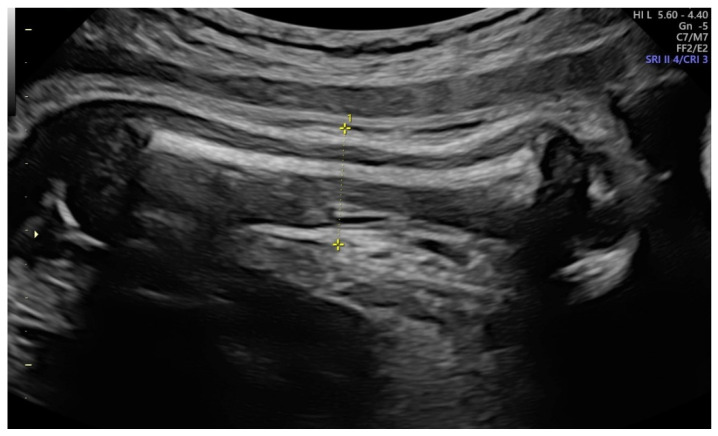
Mid-arm diameter, evaluated placing the caliper from skin to skin on the upper arm near the heart.

**Figure 2 ijerph-19-05747-f002:**
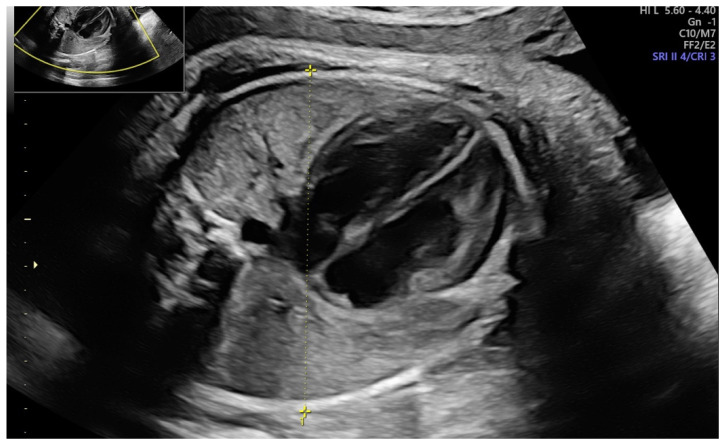
Transverse thoracic diameter (TTD) estimated by determining a transverse section of the fetal chest at the level of the heart (4-chamber view).

**Figure 3 ijerph-19-05747-f003:**
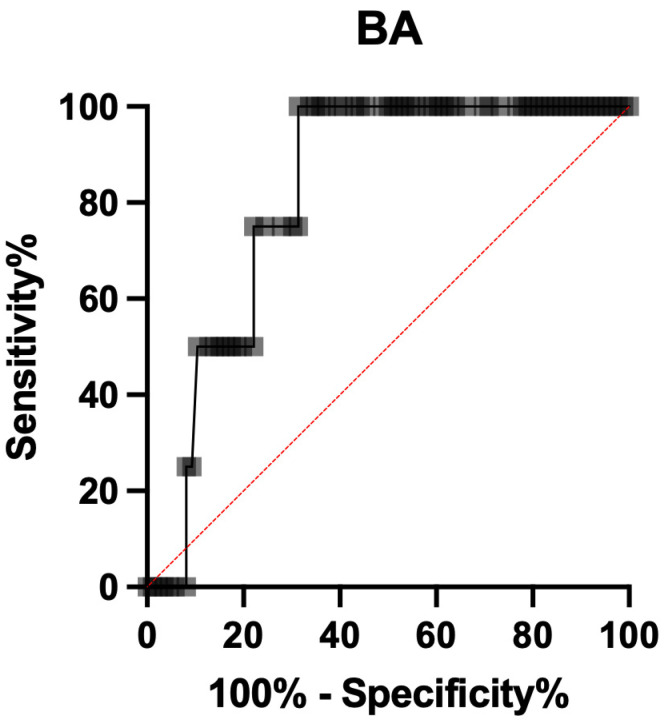
ROC curve of BA diameter for shoulder dystocia prediction.

**Figure 4 ijerph-19-05747-f004:**
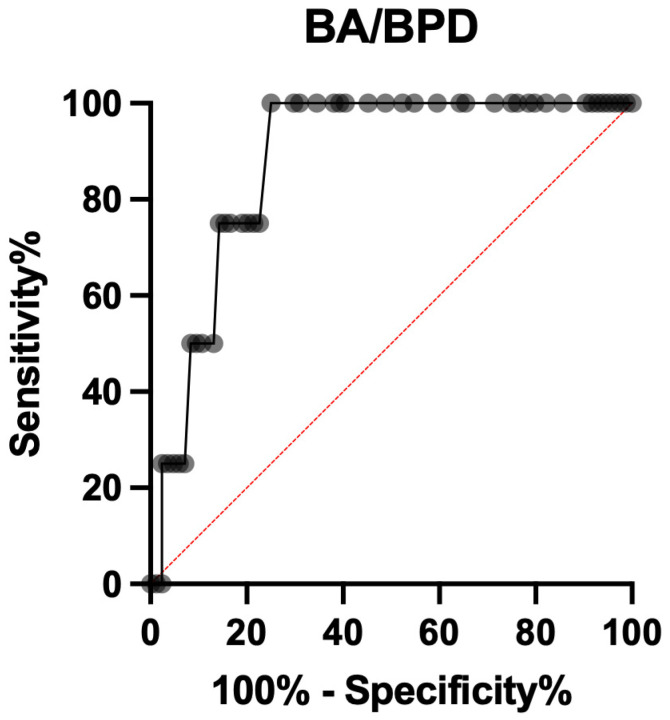
ROC curve of BA/BPD for shoulder dystocia prediction.

**Figure 5 ijerph-19-05747-f005:**
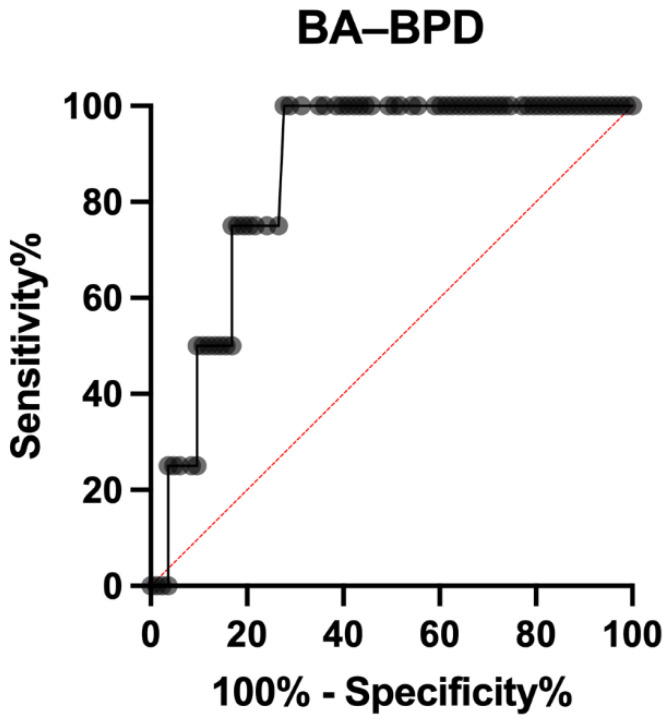
ROC curve of BA–BPD for shoulder dystocia prediction.

**Figure 6 ijerph-19-05747-f006:**
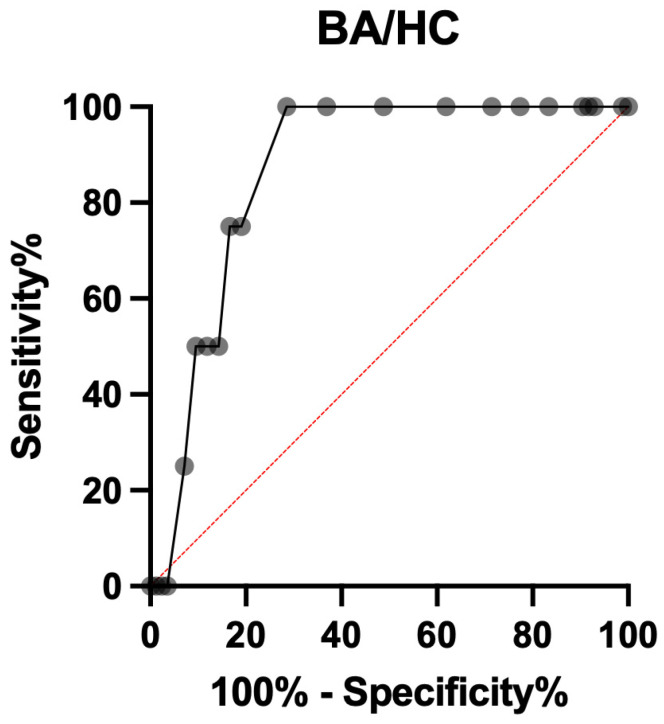
ROC curve of BA/HC for shoulder dystocia prediction.

**Table 1 ijerph-19-05747-t001:** Demographic data of the study participants (n = 90).

Age, (years)	
mean ± SD	31.67 ± 5.6
Height, (cm)	
mean ± SD	162.8 ± 5.7
Pre-gestational weight, (Kg)	
mean ± SD	78.4 ± 14.47
BMI, (Kg/m^2^)	
mean ± SD	29.65 ± 5.18
Weight gain, (Kg)	
mean ± SD	15.88 ± 17.05
Week of gestation	
mean ± SD	40.0 ± 1.1
Parity,	
mean ± SD	0.4 ± 0.7
Primigravida,	
number (%)	40 (44.4)
Multigravida,	
number (%)	60 (55.56)
Previous miscarriage,	
number (%)	24 (26.67)
Gestational diabetes,	
number (%)	11 (12.22)
Pregestational diabetes mellitus,	
number (%)	1 (1.11)
Gestational hypertension and preeclampsia,	
number ± (%)	8 (8.89)

SD, standard deviation; BMI, body mass index.

**Table 2 ijerph-19-05747-t002:** Ultrasonographic fetal measurements.

	ShD (4)	No ShD (86)	*p*-Value
Biacromial diameter, cm			0.04 *
mean	150.4	133.5	
95% CI	133.2–167.6	130.1–137.0	
BA/BPD,			0.04 *
mean	1.66	1.44	
95% CI	1.46–1.86	1.41–1.48	
BA/HC,			0.01 *
mean	0.45	0.39	
95% CI	0.40–0.49	0.38–0.40	
BA–BPD,			0.03 *
mean	60.0	41.4	
95% CI	42.4–77.6	38.2–44.6	
Estimated fetal weight, g			0.61
Mean ± SD	3273 ± 618	3448 ± 352	

SD, standard deviation; CI confidence interval; BA, biacromial diameter; BPD, biparietal diameter; HC, head circumference. * *p* < 0.05.

**Table 3 ijerph-19-05747-t003:** Delivery data of the study participants.

Mode of Delivery Number (%)	Vaginal Cesarean Section Operative Delivery	78 (86.67) 12 (13.33) 10 (11.11)
Apgar score, mean ± SD	1′ 5′	8.2 ± 0.89 9.4 ± 0.54
Birth weight, gr mean ± SD		3426 ± 32
Birth length, cm mean ± SD		51.32 ± 1.62
Cranial circumference, cm mean ± SD		34.62 ± 1.12

SD, standard deviation.

## Data Availability

The data presented in this study are available on request from the corresponding author.

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
