# Peer review of "Accuracy of Fetal Biacromial Diameter and Derived Ultrasonographic Parameters to Predict Shoulder Dystocia: A Prospective Observational Study"

_ijerph, 2022, doi:10.3390/ijerph19095747_

Round 1
Reviewer 1 Report
In my opinion the main limitation of the study is small number of cases - especially when the incidence of ShD is low (only 4 patients). Has this study in this circumstances appropriate power to asses the risk of ShD based on studied parameters? It should be statistically evaluated. The authors included only 90 patients within a year - it seems to be very small for a tertiary hospital - can you explain the reason of this? The rate of delivery by cesarean section is very low 13,3% - it may suggest that the study group is highly selected. Additional informations about patients with ShD should be provided - where they nulli or multiparous, what about their risk factors for ShD? The prediction of ShD based on ultrasound parameters should also take into account such factos, as BMI, parity and coexistence of DM. Ultrasound fetal biometric parameters might be different between fetuses of mother with and without DM, because of the different distribution of adipose tissue. I think the authors should explain these problems more detailed in discussion. Discussion section is very superficial in my opinion.
There are also some inaccuracies in the mansucript i.a.: duration time of the study in abstract is March 2021 to February 2022, in the main text it is March 2021 to January 2022; lines 92-93 there is a doubled sentence.
Reviewer 2 Report
Shoulder dystocia represents an extremely dangerous obstetric complication; therefore, the subject is of great importance in obstetrical practice, it prevention been vital. The study is an attempt to develop and evaluate a reproducible method of screening to prevent this unpredictable complication.
In the Method section is describe the technique to measure the transverse thoracic diameter and mid-arm diameter. I would find it interesting to add a demonstration image of how these diameters are measured.
I also do not understand the purpose of the citations in the phrase: "We excluded pregnant with antepartum hemorrhage [28], fetal growth restriction [29], stillbirths [30], multiple gestations [31], breech position [32,33], uterine or congenital malformations [34-36].
Discussions could be improved.
Reviewer 3 Report
Overall I am pleased with the scientific contribution made by the authors and would like to congratulate them for their idea and work. However, I would like to mention that there are too many bibliographical references for an original research paper and the discussion part is far too succinct. The discussion needs more reference to the literature. Also, it would be worthwhile to point out the applicability of this method in current practice and possibly to improve the previously validated formula, with a potential inclusion of it in protocols.
Round 2
Reviewer 1 Report
Thank You very much for Your effort to improve the manuscriptand congratulations.
The authors significantly imporved their manuscript , especially the discussion section, explained all indicated and unclear elements and now I recommend it for publishing
Reviewer 2 Report
All comments were answered appropriately